

# Brief communication: Full-field deformation measurement for uniaxial compression of sea ice by using the digital image correlation method

Anliang Wang[1], Zhijun Wei[2], Xiaodong Chen[3], Shunying Ji[3], Yu Liu[1], Longbang Qing[4]

[1]Marine Disaster Forecasting and Warning Division, National Marine Environmental Forecasting Center, Beijing, 100081, China.
[2]State Key Laboratory of Coastal and Offshore Engineering, Dalian University of Technology, Dalian, 116024, China
[3]State Key Laboratory of Structural Analysis for Industrial Equipment, Dalian University of Technology, Dalian, 116024, China
[4]School of Civil Engineering and Transportation, Hebei University of Technology, Tianjin, 300401, China.

*Correspondence to*: Longbang Qing (qing@hebut.edu.cn)

**Abstract.**

The study of the mechanical properties of sea ice benefits the parameterization of sea ice numerical models and optimization of engineering design. Deformation measurement of sea ice has been believed to be the essential foundation for the study.

However, this measurement has been proved to be difficult due to the complex and nonhomogeneous mechanical properties of sea ice. In this paper, we took advantage of DIC (digital image correlation) to obtain the full-field displacement and strain of sea ice specimen in a uniaxial compression experiment. Full-field deformations of sea ice under both vertical and horizontal loading were measured. Different mechanical behaviors such as microcrack and failure modes due to the anisotropic properties of sea ice were successfully captured. The nonuniformity and local concentration of the strain field were observed and analyzed.

Additionally, we evaluated the displacement and strain field of the specimens to verify the feasibility and accuracy of the method. This successful application provides a convenient and powerful option for the study of sea ice mechanical properties such as failure mode, nonlinear behavior and crack propagation.

## 1 Introduction

Human activities increase with more space available to be exploited (explored) in polar areas (Laliberté et al., 2016; Rabatel

et al., 2018). This trend requires better numerical models of sea ice and cold region technology to reduce the risk from the ice (Rabatel et al., 2018). Studies of the mechanical properties of sea ice have been believed to make such contributions (Timco and Weeks, 2010; Shokr and Sinha, 2015; Weiss and Dansereau, 2017). For example, the strength of flexure and compression (Ji et al., 2011), failure and fracture mode (Schulson, et al., 2006; Weiss, 2013; Lian et al., 2017), and Young's modulus and Poisson ratio (Schulson, and Duval, 2009) all have a close relationship with the parameterization of sea ice models (Hibler,

1979; Feltham, 2008; Weiss and Dansereau, 2017) and optimization of polar engineering designs (Ibrahim et al., 2007).



Deformation measurements at the laboratory scale are an essential foundation of those studies. However, it has been found to be difficult to measure the deformation due to the complex material properties of sea ice (Cole, 2001), even under the controlled conditions of a well-equipped laboratory (Sinha,1984). In particular, measurement of full-field deformation for sea ice specimens has not been reported.

In traditional ice mechanics, displacement actuators, strain gauges and extensometers are occasionally applied to measure the deformation of specimen (Schulson, and Duval, 2009; Timco and Weeks, 2010). The equipment for measuring displacement is generally installed onto the specimen surface. The operation sometimes causes local damage and thus increases the local stress concentration. Another option is to obtain the equivalent displacement of the specimen from an indenter (Schulson, and Duval, 2009; Wang et al., 2018). The loading surface of the specimen and indenter on it move together and thus have the

same displacement all the time. Therefore, the strain of the specimens can be deduced from the displacement of indenter without considering the deformation of the rig itself. This method provides only one value to represent the overall deformation of the specimen, even under the assumption that the loading system has enough stiffness compared with that of sea ice specimen, For sea ice material, the local variation in deformation cannot be ignored because of the existence of brine pockets and air bubbles inside of sea ice (Schulson, and Duval, 2009; Li et al., 2010). Therefore, full-field deformation is needed to capture

the local conditions of sea ice specimens and depict the failure characteristics during the loading process.

Fortunately, the digital image correlation (DIC) method has been developed (Sutton et al., 2009, 2016), which is suitable for deformation measurements. Based on the DIC, full-field displacement and strain can be accurately obtained by comparing the digital images of specimen surfaces for the initial and deformed states (Pan et al., 2009). This method has been widely used in many fields to obtain the full-field deformation (Sutton et al., 2016). Recently, Lian et al. (2017) used DIC to investigate the

uniaxial compressive strength and fracture mode of natural lake ice under moderate strain rates. They found that the strain rate calculated from DIC is quite different from the one deduced from actuator displacement under dynamic loading conditions (Lian et al., 2017). Based on the DIC technique, the full-field deformation is accurately deduced and the damage process of the specimen is clearly obtained at high spatiotemporal resolution (Lian et al., 2017). In fact, a similar principle has been applied to sea ice satellite images to compute the velocity and strain fields on a geophysical scale (Muckenhuber and Sandven,

2017). Nevertheless, we have not found the application of DIC in sea ice mechanical property experiments either in the laboratory or in situ. This is partly because of the complex material properties and intricate mechanical behaviors of sea ice, which make such applications more difficult.

In this paper, we attempt to apply the DIC technique to sea ice uniaxial compression experiment in situ. First, we introduce the experimental procedure and briefly interpret the DIC theory. Then the displacement and strain field of specimens are

illustrated and analyzed to certify the feasibility and accuracy of the method. To our knowledge, this is the first attempt to experimentally capture sequential full-field deformations in the mechanical properties of sea ice. This achievement will extend the ability to further explore the complex mechanical behaviors of sea ice.



## 2 Materials and methods

### 2.1 Specimens and equipment

The experiment was carried out at the Bayuquan ocean station (40°07′15.32″ N, 121°57′34.77″ E) in Liaodong Bay where there is an ice-covered season of approximately three months each year. A large ice block 1.0 m × 1.0 m was cut from a level ice sheet using a chain saw, and the thickness was approximately 30 cm. Meanwhile, we measured the salinity of seawater and collected several pieces of sea ice for salinity measurements. The type of crystal structure of the ice was columnar with column diameter of approximately 4 mm. The seawater salinity was 33 ppt at the sampling site, and the ice salinities were between 5.5

ppt and 7.4 ppt with a mean value of 6.1 ppt. The environmental temperature during our experiment was about -10℃, and the ice temperatures were between -4.5℃ and -5.6℃ with a mean value of -4.9℃. These ice blocks were finely processed into cuboid specimens with sizes of 50mm×50mm×107mm using a band saw. This size ensures that the specimens contain enough ice crystals to avoid grain boundary effects (Timco, and Weeks, 2010; Ji, et al., 2011) and meets the maximum load requirement of 3 t for our loading system.

To obtain a high-contrast speckle pattern, we first sprayed white paint uniformly on the specimen surface as the background and then sprayed black paint randomly to produce speckles on the white surface. The specimen surface tended to be flat and smooth after the white paint was sprayed at least four times. Every spraying required an interval of ten minutes. Half an hour or more after applying the white paint, the black paint was sprayed onto the white surface. During spray painting, we maintained the outlet of black paint at a specific distance from the white surface, which was greater than 40 cm in our

experiment. This control ensured that the mist of black paint randomly fell onto the white surface and thus resulted in a random gray intensity pattern on the surface. Note that touching on the sprayed surface of the specimen was always forbidden during the experiment to prevent the contamination of the speckle pattern. Sitting for at least two hours after being spray painted, the specimens were ready for the compression experiment.

The experimental system mainly consisted of an optical image acquisition device and loading system. The CCD (charge

coupled device) camera and high-intensity light source were the major components of the former. The camera was Basler acA 1600-20gm with a resolution of 1200 pixels ×1600 pixels. Approximately, this resolution assured specimen surface of 500 pixels ×1070 pixels, which was tantamount to 0.1 ± 0.005 mm per pixel for each frame. The camera was placed parallel to the specimen surface, and the distance between them was kept properly long, greater than 2.5 m in our experiment. This arrangement could alleviate the influence of the out-of-plane deformation (Pan et al., 2009; Sutton et al, 2009). The camera

had a frequency of 20 fps (frames per second) to trace the deformation of the specimen surface. For the loading system, the load was applied from bottom to top by a servomotor, at the bottom of the apparatus, which could supply the maximum force requirement of 3 t with a constant speed of 0.001-0.8 mm/s. Here, we used a low loading speed of 0.05 mm/s. Therefore, the frequency of the CCD camera was sufficient to capture the development of the full-field deformations. The time history of the load and displacement of the indenter were simultaneously recorded by the loading system.

### 2.2 Methods



The DIC method computes deformation information by matching the speckles of the specimen surface before and after loading stages. Generally, the equilateral grids are virtually assigned on the specimen region of interest. The center of the subset carries the displacement information, as illustrated in figure 1 (b-c). The subset generally consists of an area of (M+1) pixel×(M+1) pixel. After assignment of the subsets, the appropriate matching method for the centers between the initial image and deformed image could be determined. Considering the robust noise-proof performance, we used the following correlation criterion (Pan et al., 2009):

$$C = \sum_{i=-M}^{i=M} \sum_{j=-M}^{j=M} \left( \frac{f_{ij}-\overline{f}}{\Delta f} - \frac{g_{ij}-\overline{g}}{\Delta g} \right)^2 \qquad (1)$$

where $f_{ij}$ and $g_{ij}$ are gray-level functions for initial image and deformed image, respectively; $\overline{f}$ and $\Delta f$ are defined as $\overline{f} = \frac{1}{(2M+1)^2} \sum_{i=-M}^{i=M} \sum_{j=-M}^{j=M} (f_{ij})$ and $\Delta f = \sqrt{\sum_{i=-M}^{i=M} \sum_{j=-M}^{j=M} (f_{ij} - \overline{f})^2}$ in the initial image; the same definitions of $\overline{g}$ and $\Delta g$ are calculated depending on $g_{ij}$ in the deformed image. When the correlation coefficient $C$ reaches extrema, the center point of the subset in the initial image is matched to the deformed image. Figure 1 (b-c) illustrates that the displacement $\vec{u}$ of the center point is computed according to the matching information.

Based on the displacement information of the center point at the initial subset, we can further obtain the displacement field for all points. Under the assumption that the deformation is continuous, all the neighboring points in the deformed image remain in the same order in the deformed image. Therefore, all the coordinates around the center point in figure 2 (b) can be mapped to the points of the deformed subset in figure 2 (c) according to the shape function, similar to the finite element method. Finally, we obtained the displacement and strain field for the entire surface of the specimen.

## 3 Results

Full-field deformations of sea ice under both vertical and horizontal loading directions were measured. These two loading directions are defined in reference to the ice crystal orientation, which is parallel to the vertical loading direction and perpendicular to the horizontal loading direction. Basically, loading direction can influence the mechanical behaviors due to the anisotropic properties of sea ice (Timco, et al., 2010). In our experiment, the DIC was able to capture these mechanical characteristics by full-field deformation.

The displacement and strain fields for the *x*- and *y*-directions were obtained based on the images of the ice surface, as shown in figure 1 (d). The coordinates were defined according to figure 1 (b-c). In this experiment, minus and plus signs represented compressive and tensile deformation, respectively. We carried out seven groups of experiments, of which three were subjected to vertical loading. We selected a 50 mm × 50 mm × 105 mm region from the digital images of the specimen surface for the DIC analysis. Figure 2 shows the evolution of the strain fields for different loading stages. The full-field strain showed nonuniformity both under vertical and horizontal loading, and the localization appears to be significant in figure 2 (a-b). Even during the elastic stages, which were identified by the displacement-load curve of the indenter in figure 2 (c), the nonuniformity



remained the same as that in the following plastic stage. For example, figure 2(a) exhibits strain fields for the *x*- and *y*-directions corresponding to four time points (H$_1$, H$_2$, H$_3$ and H$_4$) that are marked on the displacement-load curve in figure 2 (c), and point H$_3$ is instant when the maximum load occurs. The first two columns obviously presented irregular strain distributions, although they were in the elastic stage. In particular, some bottom parts of the specimens experienced larger strain values than other

parts. This trend became more observable as the load applied on the specimen increased. A more perceptible trend occurred in the strain field of the *x*-direction, and the relatively large values were apparently concentrated in the bottom parts. This coincidence between the *x*- and *y*-direction strain fields may mean local failure began at some bottom parts of the specimens, but the time points of H$_1$ and H$_2$ were still in the elastic stage according to the displacement-load curve in figure 2 (c). Unlike the typical metal displacement-load curve, sea ice had no clear yield points in figure 2 (c). Here, we took points located in the

linear segment of the displacement-load curve for elastic analysis. As we predicted, the specimens under vertical loading failed in splitting failure mode, as shown in figure 2 (d). The final crack distribution in figure 2 (d) further corroborated our speculation about the initial failure. The local damage configuration (the dashed-line circle) and vertical crack distribution in figure 2 (d) were accurately captured by the corresponding strain fields of H$_3$ and H$_4$ in figure 2 (a). Some types of strain concentration frequently happened during our experiment, but the overall displacement-load curves in figure 2 (c) hardly

represented its evolution.

From figure 2 (a-b), we observed two different failure modes and strain distributions caused by different loading directions. The strain fields of the *y*-direction in figure 2 (a) demonstrate the tendency of layered distribution parallel to the direction of ice crystal orientation, while figure 2 (b) does not exhibit this regularity. However, figure 2 (b) shows that the propagation direction of the main crack was parallel to crystal orientation. One interesting phenomenon was the recovery that some parts

underwent relatively large strain in the *x*-direction and then alternately took low strain during the elastic stage experienced, such as the two squares within the white rectangle in figure 2 (b). However, once the fracture took shape at the maximum loads at V$_3$ and H$_3$ in figure 2 (a-b), the subsequent fractures could propagate based on that shape and result in an irreversible process. Compared with the picture at the maximum loads V$_3$ and H$_3$, the fractures were notably expanded for the strain fields of *x*- and *y*-directions at the point of V$_4$ and H$_4$. Obviously, sea ice stiffness parallel to the crystal orientation tended to be higher than

that perpendicular to the ice crystal orientation. In turn, the strength of vertical loading (V$_3$) was greater than that of horizontal loading (H$_3$), as shown in figure 2 (c). Figure 2 (d) and (f) show ductile and splitting failure for two specimens. We noticed that even at the same indenter velocity, the different failure modes existed for two specimens because of the different orientations of the ice crystal. All these strain characteristics of the two failure modes and the local fracture propagation were exactly captured by the DIC.

**4 Discussion**

For this application of DIC, producing a high quality speckle pattern on the specimen surface becomes to some extent the most challenging operation. The specimen surface needs to be covered with random speckle patterns (Pan et al., 2009), which are in fact the carriers of deformation information during the loading process. Unfortunately, there is no natural texture on the





surface of sea ice specimen to be used. Therefore, the high quality of the speckle pattern is an essential prerequisite for our experiment. In practice, the speckle patterns may show distinctly different intensity distribution characteristics and have a significant influence on DIC measurements. Our major challenge came from the natural properties of sea ice that entrap salt brine pockets and air bubbles. As a result, the surface of the cut specimens retained some faults, which could not be polished

off the same as those on the surface of lake ice (Lian et al., 2017). We artificially made the speckle pattern by spraying with white and black paints to overcome this drawback. Then we applied the mean intensity gradient to the quality assessment of the speckle pattern. This method is straightforward and uses easy-to-calculate global parameters to assess the quality of the speckle pattern. The mean intensity gradient is given as follows (Pan et al., 2010):

$$\delta_f = \sum_i^W \sum_j^H |\nabla f(x_{ij})| / (W \times H) \tag{2}$$

where $W$ and $H$ are the width and height of the specimen in units of pixels, $|\nabla f(x_{ij})| = \sqrt{f_x^2 + f_y^2}$ is the modulus of the local intensity gradient vector and $f_x$, $f_y$ are intensity derivatives at $x_{ij}$ with respect to the $x$- and $y$-directions. Here, we took advantage of the Sobel operator to compute the intensity derivatives. The mean intensity gradients of the initial images for specimens of figure 3 (b) and (c) were 172.01 and 182.20, respectively. All the values of specimens for this experiment were between 146.11 and 182.20 with an average of 164.08. These values suggest a high quality speckle pattern and subsequently

a high accuracy subset match in the DIC (Pan et al., 2010). From figure 3 (a), we found that the gray-level histogram for the speckle pattern tends to be a random distribution. Here, we took advantage of false-color to enhance contrast of the gray-level histogram. The randomness also indicates that we achieved a high quality speckle pattern on the specimen surface for the measurements.

Our selection of the subset size was 35 pixels×35 pixels, as shown in figure 3 (a). This took consideration of the large,

complex and nonhomogeneous deformation of sea ice specimens. On the one hand, the size of this subset was large enough to contain distinctive intensity variations so that every subset could own a unique speckle pattern to benefit matching accuracy; on the other hand, this selection reduced the additional systematic errors in measured displacements. From figure 3 (a), we can see that every subset makes itself different from the others by comprising enough speckles to possess its own unique gray-level distribution.

In our experiments, strain concentrations in localized parts were often observed from the strain fields of specimens. This is most likely because of the existence of sea ice defaults. These defaults are generally caused by air bubbles and salt brine pockets that are entrapped during the growth process of sea ice (Shokr and Sinha, 2015). In nature, they are randomly distributed inside the sea ice. Therefore, the strain concentration caused by the defaults could hardly be avoided through just the artificial handling of the specimen surface. This random distribution of default, to some degree, defines the difference in

mechanical properties of sea ice from lake ice and other materials (Schulson, 1999; Cole, 2001; Shokr and Sinha, 2015; Weiss and Dansereau, 2017)). Furthermore, the strength, failure mode and nonlinear mechanical behavior of sea ice are all related to these defaults and their random distribution (Schulson, 1999; Cole, 2001; Li et al., 2011). The strain concentration was exactly captured and further supported the feasibility of the DIC to study the mechanical properties of sea ice material.



Additionally, we compared the displacement with that obtained from the indenter to evaluate the results of DIC. In our experiment, the indenter and the bottom of the specimen shown in figure 1 (e) should have the same displacement. The displacements of the indenter were recorded by the loading system during our experiment. The bottom displacements can be derived from the digital images based on the DIC. Here, we averaged the displacements of two bottom lines of pixels as the

bottom displacements of the specimen. Figure 3 (b) demonstrates that those two sources of displacement agree well during the whole loading history. The red line represents the displacements of the indenter corresponding to the loading history. Even across the point of the maximum load where some nonplanar strain may subsequently occur on the surface of the specimen, the points in figure 3 (b) are still close to the red line. This coincidence suggests that our control of the distance between the CCD camera and specimen surface has a beneficial effect on reducing the influence of out-of-plane strain on the measurement

of the full-field deformation. Nevertheless, when the loading process exceeds 100 s, the deviation seems to be evident. This is mainly because of the large fracture and damage of the specimen. In addition, the deviation may partly come from the selection of the analyzed region whose bottom line is one centimeter above the bottom line of the specimen in the initial images. However, the reliability of DIC in the deformation measurement of sea ice is confirmed by the comparison.

### 5 Conclusion

In conclusion, we successfully applied the DIC technique to measure the deformation of sea ice in a uniaxial compression experiment. The full-field deformations are obtained and the ability of DIC to capture the strain concentration and failure modes is confirmed. The local and global characteristics of ductile and splitting failures are accurately reflected by the strain fields of the $x$- and $y$-directions. In addition, the gray-level distribution and comparison of displacements from the indenter and DIC are assessed. The results corroborates each other and bolster confidence in the reliability of the method, the quality of

speckle pattern and accuracy of the full-field deformation measurement in the experiment. Additionally, methods similar to DIC could be extended to retrieve the motion and deformation of sea ice on a geophysical scale based on high quality of satellite images (Marsan et al., 2004; Komarov and Barber, 2014; Muckenhuber and Sandven, 2017). DIC method provides a convenient and powerful tool for the study of sea ice mechanical properties such as failure mode, nonlinear behavior and crack propagation.


*Acknowledgements*. This work is supported by the National Key Research and Development Program of China [2016YFC1401500], the National Natural Science Foundation of China [41506109, 41676189 and 11602051] and the China Postdoctoral Science and Foundation [2016M591433]. The authors thank Guorui Cao for many suggestions on the design of the experiment.

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



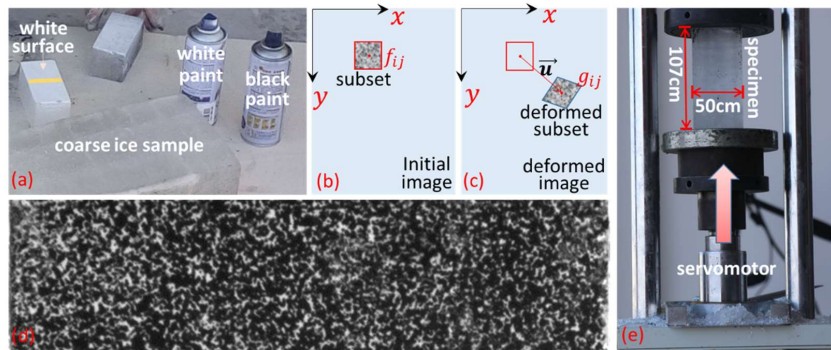

5    **Figure 1:** Specimen preparation, definition of coordinates and loading system. (a) White/black speckles from spraying the oil paints on the specimen surfaces; (b-c) the matching process between the initial subset and deformed subset; (d) magnified part of the prepared specimen corresponding to the yellow color in (a); (e) the load applied the specimen from bottom to top.





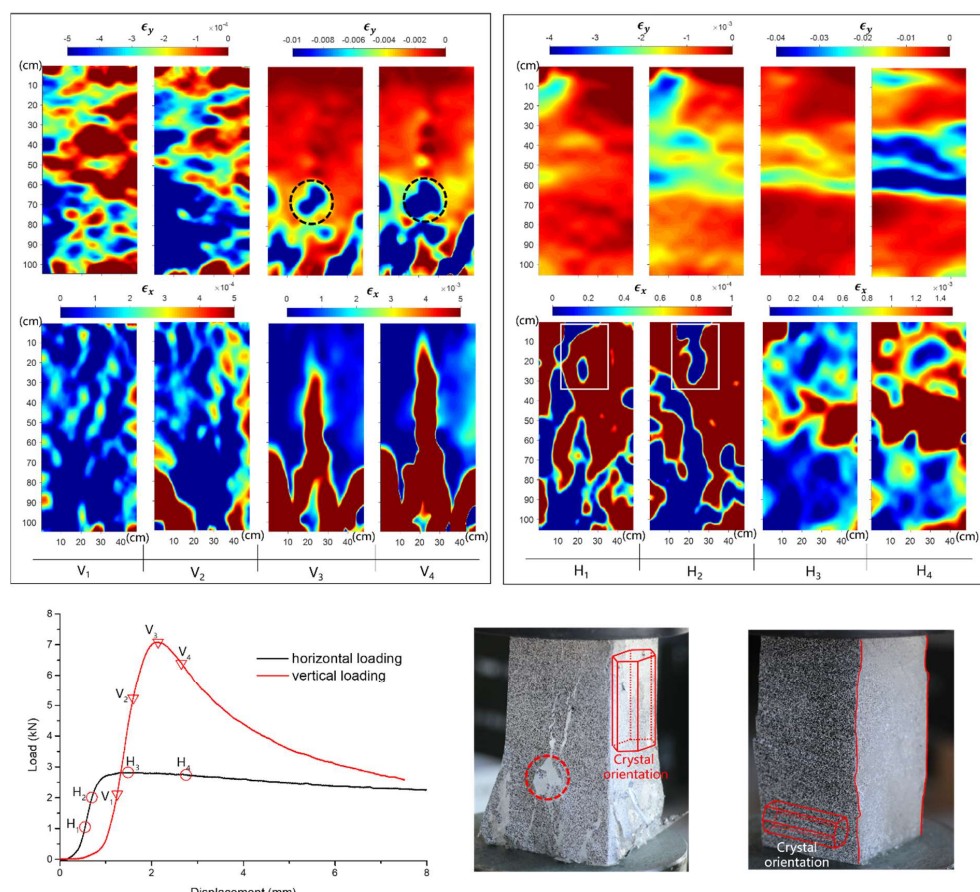

**Figure 2:** The evolution of strain fields in the uniaxial compression experiment. The strain fields of specimens (a) and (b) with respect to the vertical and horizontal loading; the $x$- and $y$-direction strain fields are located in the first and second rows. The columns of $V_i$ and $H_i$ ($i$ = 1, … ,4) correspond to the time points labeled in the displacement-load curves of (c). The failure pictures (d) and (e) for the specimens (a) and (b). Note that the same color bar is shared with the two neighboring strain fields in (a) and (b) for the enhancement of the color contrast.





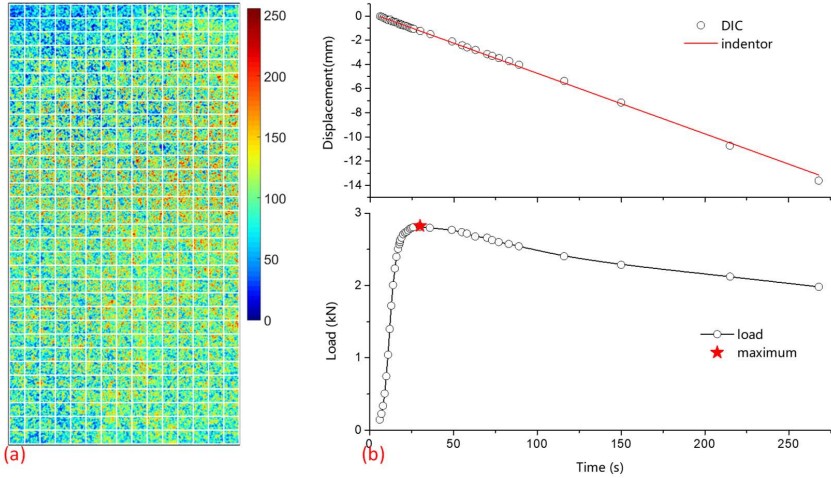

5 **Figure 3:** (a) Distribution of gray-level histograms of subset grids in false-color on the surface of a spray painted specimen and (b) comparison of displacements from the indenter and DIC method during the loading process.