# Peer review of "Brief communication: Full-field deformation measurement for uniaxial compression of sea ice by using the digital image correlation method"

_The Cryosphere, 2018_

## Referee Comment (RC1) · Knut Høyland (Referee) · 14 Feb 2019

Dear Authors I find you paper interesting and well written. It demonstrates the use a (to me at least) novel technique that can be used to monitor the strain field in field testing of ice. I do not understand the DIC technique, but assume that you have described it and used it correctly. Page 1, Line 29. I don't think there is a close and precise relationship between large-scale models and small-scale properties. The same goes for ice action on ships and structures. The ice properties are of course essential, but, the current models are not detailed enough to make real use of small-scale mechanical properties of ice. Page 3, Line 14. How many Vertical and how many Horizontal samples did you

test? Page 4. I assume you find the displacement filed from the DIC method. Which strain definition did you use? Page 5, Line 4. The samples most probably responded with delayed elastic and perhaps even creep early in the test. It is to simple to claim that V1, V2, H1 and H2 were in purely elastic range. Page 5. What about plotting the von-Mises strain, or the shear strains also?

---

## Editor Comment (EC1) · Christian Haas (Editor) · 30 Mar 2019

Dear authors,

I am happy to accept your manuscript for publication after careful consideration of the comments of reviewer 1. Please provide replies to those comments and modify your manuscript accordingly. In addition, I agree with the reviewer that the applications of your method to larger scales are far fetched. Please could you discuss more carefully what the applicability and relevance of your method really is, on the small scales of your sample and experiment. What can we learn from such small scale experiments, and how do the small scale material properties of sea ice affect the representativity and

reproducability of the method. It would also be good if you could provide a little more information on the image processing involved with the method. Please add labels a-e to figure 2. Thank you and best regards

Christian Haas Editor, TC

---

## Author Comment (AC1) · 25 Apr 2019

**Response to Referee 1 (Knut Høyland)**

1. Dear Authors, I find your paper interesting and well written. It demonstrates the use a (to me at least) novel technique that can be used to monitor the strain field in field testing of ice. I do not understand the DIC technique, but assume that you have described it and used it correctly.

**Response:** Thank you for referee's comments. The recognition surely encourages us to further apply the DIC method to investigating the mechanical properties of sea ice.

As far as we know, DIC has been widely used to get the full-field deformation and then monitor mechanical behaviors in many research fields. Its accuracy and efficiency have been proved by many types of application (Pan et al., 2009; Sutton et al., 2016). However, DIC is a relatively fresh technique in the research of sea ice mechanical properties. In our experiment, the coincidence of displacements obtained by the indenter and DIC method also verifies the accuracy of DIC method in measuring the deformation of sea ice. Furthermore, DIC method shows excellent ability to in situ test with the easy-to-perform advantage.

However, sea ice is a heterogeneous material including crystalline ice, bubble and brine (Weiss et al., 2017), which have a negative effect on the speckle images for the measurement of sea ice deformation. So we suggest that much more attention should be paid to produce high-quality speckles on sea ice surface especially in the low temperature environment. This is an essential prerequisite for acquiring the speckle images with high quality. All the other aspects such as image acquisition and data analysis are almost the same with the standard process of DIC technique.

2. Page 1, Line 29. I don't think there is a close and precise relationship between large-scale models and small-scale properties. The same goes for ice action on ships and structures. The ice properties are of course essential, but, the current models are not detailed enough to make real use of small-scale mechanical properties of ice.

**Response:** Thank you for referee's suggestion. We changed the sentence on page 1, line 29 according to referee's suggestion. Moreover, in order to avoid confusion, we deleted the contents about the application of DIC in

geophysical scale in *Conclusion*. Potential application of DIC to the study of sea ice mechanical properties was detailed as well.

3. Page 3, Line 14. How many Vertical and how many Horizontal samples did you test?

**Response:** We totally carried out seven groups of test and three of them were horizontal samples. This is declared on page 4, line 27. However, the failure processes of three samples were not captured by the DIC method, because the surface with white/black speckles was not the right surface where the failure process happened. Therefore, if possible, white/black speckles should be produced on two orthogonal facets and the fracture will be more likely to happen on one of them. Moreover, two cameras are needed to simultaneously capture the characteristics of the two facets. Alternatively, the loading surface of sample may be processed into rectangle such as 50mm×60mm to reduce the randomness of failure surface. In our experiment, the size of sample was 50mm × 50mm × 107mm, so the failure surface may happen on any facets.

4. Page 4. I assume you find the displacement filed from the DIC method. Which strain definition did you use?

**Response:** The nominal strain was adopted in our paper. We got the full-field deformations by comparing all deformed images with the initial image which is unique for each test. This computation agrees with the definition of the nominal strain. We declared this on page 4, line 25.

5. Page 5, Line 4. The samples most probably responded with delayed elastic and perhaps even creep early in the test. It is too simple to claim that V1, V2, H1 and H2 were in purely elastic range.

**Response:** We agree that V1, V2, H1 and H2 were not in purely elastic stages, so we replaced *elastic stages* with *early stages before yielding* for this updated version.

6. Page 5. What about plotting the von-Mises strain, or the shear strains also?

**Response:** Many thanks for referee's advice. This valuable advice will make the failure characteristics more observable in figure 2. Therefore, we added one row of sub-figures of shear strain ($\varepsilon_{xy}$) to figure 2(a) and (b),

respectively. Several explanations were accordingly given in the paper. Then, we updated figure 2 on page 11 and the corresponding explanation was given on page 5, lines 28-30.

**References:**

Pan, B., Qian, K., Xie, H., and Asundi, A.: Two-dimensional digital image correlation for in-plane displacement and strain measurement: a review, Measurement Science & Technology, 20 (6), 062001, doi:10.1088/0957-0233/20/6/062001, 2009.

Sutton, M. A., Matta, F., Rizos, D., Ghorbani, R., Rajan, S., Mollenhauer, D., H., Schreier, H. W., and Lasprilla A. O.: Recent progress in digital image correlation: background and developments since the 2013 W M Murray lecture, Experimental Mechanics, 57(1), 1-30, doi: 10.1007/s11340-016-0233-3, 2016.

Weiss, J., and Dansereau, V.: Linking scales in sea ice mechanics. Philosophical Transaction Royal Society A, 375, 20150352, doi: 10.1098/rsta.2015.0352, 2017.

---

## Author Comment (AC2) · 25 Apr 2019

**Response to Editor (Christian Haas)**

1. I am happy to accept your manuscript for publication after careful consideration of the comments of reviewer 1. Please provide replies to those comments and modify your manuscript accordingly.

**Response:** Thanks for Editor's advices. The comments from Reviewer 1 have all been carefully considered and some parts of our paper have subsequently modified. We do our best to make this updated version to entail the requirement of TC publication.

2. In addition, I agree with the reviewer that the applications of your method to larger scales are far fetched.

**Response:** Thanks for Editor's advices. We are sorry for this confusion. From the perspective of image processing, two types of image share many commons although they come from different sources and have different spatial scales. Especially, the similar matching technique was used to extract the displacement information of sea ice both for experiment images (Pan et al., 2009; Sutton et al., 2016) and satellite images (Komarov and Barber, 2014). However, in order to avoid misunderstanding we deleted one sentence on page 7, lines 28-30.

3. Please could you discuss more carefully what the applicability and relevance of your method really is, on the small scales of your sample and experiment. What can we learn from such small scale experiments, and how do the small scale material properties of sea ice affect the representativity and reproducability of the method.

**Response:** Sea ice generally performs complex behaviors under the external force because of its multiphase medium which is composed of ice crystal, bubble and brine (Weiss et al., 2017). This complexity makes the measurement of deformation difficult and further influences the determination of mechanical parameters and analysis of failure characteristics. In order to overcome the difficulty, we took advantage of the digital image correlation (DIC) to get the full-field deformation of the samples for the whole process of sea ice mechanical

experiment. In the future, based on the full-field deformation, the parameter identification method (integrated digital image correlation (IDIC)) (Roux et al., 2006; Leclerc et al., 2009), can be incorporated to improve the accuracy of the mechanical properties of sea ice such as Young's modulus and fracture toughness. This combination of the experiment with numerical simulation should facilitate the illustration of the fracture mode. In order to make these potent application more clear, we modified some parts of *Conclusion*.

4. It would also be good if you could provide a little more information on the image processing involved with the method.

**Response:**

[Figure]

Figure 1 flowchart of DIC method

The process flowchart of DIC method is shown in figure 1. The detailed steps are as follows:

(1) Capture speckle images before and after deformation;

(2) Draw continuous analysis region(s) and set DIC parameters, such as subset radius and subset spacing;

(3) Perform initial guess and nonlinear optimization to obtain the whole displacement field;

(4) Smooth the displacement field and then the stains can be obtained by solving the gradients of displacements.

In practice, there exist open sources on Github that can be used to carry out the computation of DIC such as ncorr_2D_matlab (https://github.com/justinblaber/ncorr_2D_matlab) and DICe (https://github.com/dicengine/dice) that are followed by some manual files. Basically, those resources are good option to apply DIC to study sea ice mechanical properties.

5. Please add labels a-e to figure 2.

**Response:** We are sorry for this ignorance. All the labels have been added in figure 2 in this updated version. By the way, figure 2 has also been updated according to the referee's suggestion. One row for shear strain fields was added to figure 2 (a) and (b), respectively.

**References:**

Pan, B., Qian, K., Xie, H., and Asundi, A.: Two-dimensional digital image correlation for in-plane displacement and strain measurement: a review, Measurement Science & Technology, 20 (6), 062001, doi:10.1088/0957-0233/20/6/062001, 2009.

Sutton, M. A., Matta, F., Rizos, D., Ghorbani, R., Rajan, S., Mollenhauer, D., H., Schreier, H. W., and Lasprilla A. O.: Recent progress in digital image correlation: background and developments since the 2013 W M Murray lecture, Experimental Mechanics, 57(1), 1-30, doi: 10.1007/s11340-016-0233-3, 2016.

Komarov, A. S., and Barber, D. G.: Sea ice motion tracking from sequential dual-polarization RADARSAT-2 images, IEEE Transactions on Geoscience and Remote Sensing, 52(1), 121-136, doi:10.1109/TGRS.2012.2236845, 2014.

Leclerc, H., Périé, J N., Roux, S. and Hild, F.: Integrated digital image correlation for the identification of mechanical properties, MIRAGE 2009: Computer Vision/Computer Graphics Collaboration Techniques, 5496, 161-171, 2009.

Roux, S., and Hild, F. Stress intensity factor measurements from digital image correlation: post-processing and integrated approaches, International Journal of Fracture, 140, 141-157, doi:10.1007/s10704-006-6631-2, 2006.

Weiss, J., and Dansereau, V.: Linking scales in sea ice mechanics. Philosophical Transaction Royal Society A, 375, 20150352, doi: 10.1098/rsta.2015.0352, 2017.

---

## Author Comment (AC3) · 29 Apr 2019

1. General response: All the modifications this version are based on the clean version which we submitted last time.

2. Editor: thank you for revising your manuscript. It is much improved and almost ready for publication. Response: Thank you for editor's recognition. We are very happy to reach such level for publication in TC.

3. Editor: However, before we proceed I would like to ask you to include more material from your good responses in the manuscript. For example, please add the information

you have provided about the number of tests in response to the reviewer's comments number 3. Response: According to editor's suggestion and reviewer's comments, we changed the page 5, line 6 and added one sentence on page 5, lines 7-8.

4. Editor: Similarly, it may be possible to include the information from other replies in the manuscript more clearly? Response: According to editor's suggestion, we added one subtitle "2.3 Image processing" on page 4 and one figure "Figure 2: Flowchart of sea ice image processing for DIC method" on page 11. Following that, some figure numbers were changed for this version.

5. Editor: Please provide a short reply and an indication of the changes you will have made in response to my suggestions. Response: We made the indication of the changes for this version and the short replies were given according to the modifications.

Please also note the supplement to this comment:
https://www.the-cryosphere-discuss.net/tc-2018-263/tc-2018-263-AC3-supplement.pdf

**Supplement:**

[revised manuscript text omitted]